# Techno-economic assessment and innovative production of nutrient-rich jam, jelly, and pickle from *Sonneratia apetala* fruit

Md. Ripaj Uddin[1]*, Md. Abu Bakar Siddique[1], Shahnaz Sultana[1], Umme Hafsa Bithi[2], Nahida Akter[1], Abubakr M. Idris[3], Muhammad Abdullah Al Mansur[1], AHM Shofiul Islam Molla Jamal[1], Mayeen Uddin Khandaker[4,5,6]

**1** Institute of National Analytical Research and Service (INARS), Bangladesh Council of Scientific and Industrial Research (BCSIR), Dhaka, Bangladesh, **2** Institute of Food Science and Technology (IFST), Bangladesh Council of Scientific and Industrial Research (BCSIR), Dhaka, Bangladesh, **3** Department of Chemistry, King Khalid University, College of Science, Abha, Saudi Arabia, **4** Centre for Applied Physics and Radiation Technologies, School of Engineering and Technology, Sunway University, Subang Jaya, Selangor, Malaysia, **5** Faculty of Graduate Studies, Daffodil International University, Daffodil Smart City, Birulia, Savar, Dhaka, Bangladesh, **6** Department of Physics, College of Science, Korea University, Seongbuk-gu, Seoul, Republic of Korea

* ripajuddin-inars@bcsir.gov.bd, md.ripajuddin@gmail.com

**Data Availability Statement:** All data are available in the manuscript.

## Abstract

*Sonneratia apetala*, a nutrient-rich mangrove fruit, presents an opportunity for innovative food product development, offering potential health benefits and economic value through the creation of jam, jelly, and pickle. This innovative invention reveals the nutritional content including vitamins and minerals of *Sonneratia apetala* jam, jelly, and pickles from Nijhum Dwip in Hatiya Upazila, Noakhali District. These products contain Na, Mg, K, Ca, Mn, Fe, Cu, and Zn, which are essential for human nutrition. The texture and sensory qualities of the products depend on their Total Soluble Solids (TSS), acidity, moisture, pH, and total sugar content, with each parameter receiving an average score of 7 to 8 out of 9 (hedonic scale). Trace amounts of Cd, Cr, Pb, and Hg were found to be significantly below the safe consumption limits. $F^-$, $Cl^-$, $SO_4^{2-}$, soluble and total $PO_4^{3-}$ concentrations were also below safety thresholds. The moisture, ash, protein, fat, fiber, pectin, sugar, carbohydrate, and caloric values highlight the dietary benefits and energy content of these products. The products exhibited higher levels of vitamin C and minerals compared to other citrus fruits. All tested parameters met safe consumption standards, ensuring product safety. These products underwent testing for Heterotrophic Bacterial Count to guarantee their safety. A one-year shelf life is ensured by conducting quarterly storage data checks and organoleptic tests by a 10-member jury panel. The one-way ANOVA test for sensory analysis and shelf life detection indicates statistically significant results. These products help mitigate nutrient deficiencies and promote health by regulating the diet. Applying this technology in grassroots jam, jelly, and pickle production could potentially boost the local economy by approximately $10,000 annually through the creation of small industries among the coastal population.

**Funding:** The authors extend their appreciation to the Deanship of Scientific Research at King Khalid University for funding this work through a large group research project under grant number (R.G. P.2/219/45). The funders had no role in the study design, data collection and analysis, decision to publish, or preparation of the manuscript.

**Competing interests:** The authors declare no competing interests.

## 1. Introduction

The mangrove tree *Sonneratia apetala*, also known locally as 'Keora', thrives in the Sundarbans and lower Gangetic Delta coastal belt. This species is found across deltaic regions in Bangladesh, China, India, Malaysia, Myanmar, Papua New Guinea, Sri Lanka, and Africa [1]. The tree bears seasonal fruit from August to October, coinciding with periods of heavy rain that lower water salinity and pH. The mature fruit is yellow and rich, while the young fruit is green; both stages of the fruit possess a unique, cheese-like acidic taste and are rich in Vitamin C. The fruit is not only a dietary staple for many islanders in Bangladesh, India, Africa, Malaysia, Java, and Myanmar but also a preferred food source for deer in the Sundarbans [2]. Beyond its nutritional value, the *Sonneratia apetala*, is valued for its medicinal properties. The aerial parts of the plant contain beneficial compounds such as terpenoids, steroids, alkaloids, and polysaccharides. Scientific research has revealed that the methanol extract of the fruit exhibits a range of therapeutic properties, including anti-diabetic, antioxidant, antimicrobial, anti-HIV, and antimicrobial activities [3]. Similarly, the ethanol extract has been found to be cytotoxic, anthelmintic, antidiarrheal, and analgesic [4]. The high concentration of polyphenols in these fruits suggests their potential to aid in combating serious diseases, underlining the importance of further research into their benefits.

Jam is typically made by boiling crushed or chopped fruits with sugar until thickened [5]. The fruit may need to be washed, peeled, and then crushed or chopped before cooking. Jelly is a fruit spread that is made by heating fruit juice, sugar, and pectin until it forms a solid, translucent gel [6]. Unlike jam, jelly does not contain fruit pulp or particles, resulting in a smooth and firm texture. It is transparent and allows light to pass through. The main ingredients in jelly are fruit juice, sugar, and pectin, with lemon juice often added for flavour [7]. Pickled fruits, on the other hand, are preserved in brine or vinegar with spices, herbs, or flavorings [8]. The main ingredients for pickling fruits include vinegar (white or cider), water, salt, sugar, and various spices or herbs [9]. Traditional jams, jellies, and pickles often contain high levels of sugar, chemicals, and preservatives, resulting in products with poor nutritional value, shorter shelf life, and potential health risks [10]. Furthermore, standard manufacturing methods may lead to the loss of vital nutrients during processing. As a result, there is a growing interest in natural, minimally additive, and sustainable alternatives. *S. apetala* fruit stands out as a healthy food alternative due to its high antioxidant, vitamin, and dietary fiber content. Products such as *S. apetala* fruit pulp jelly, jam, and pickles offer more vitamin C and trace components compared to other citrus fruit products [11].

Jam, jelly, and pickles can be preserved using various recipes and procedures, and they can be stored in sealed jars at room temperature, in the refrigerator, or frozen, depending on the specific method. To prevent spoilage, it's essential to follow canning instructions when storing jam, jelly, and pickles at room temperature. Manufacturing of jam, jelly, and pickles is prevalent in many countries and is closely linked to domestic cooking, preserving seasonal fruits, and traditional cuisine. Jam and jelly are commonly spread on bread, toast, scones, and biscuits, and they can also be used to fill pastries, cakes, cookies, and sandwiches. Jam can even be incorporated into sauces, marinades, or glazes for savory dishes, while jelly can enhance the flavor of pastries, pancakes, waffles, ice cream, cakes, cookies, or meats and vegetables. Pickles are often served alongside sandwiches, burgers, hot dogs, and deli meats. They can also be diced and added to salads, relishes, and appetizer trays, providing a crisp and tasty snack option on their own.

This study introduces cutting-edge technology and formulations for producing jams, jellies, and pickles that align with consumer preferences and sustainability standards. The research investigates the feasibility of using *S. apetala* fruits as a consistent source for residents in deltaic

regions, aiming to create low-fat, nutrient-rich products. The study will assess the nutritional value, perform proximate and toxicity analyses, and compare the results with standard consumer safety limits for *S. apetala*-derived jams, jellies, and pickles. We plan to estimate the content of vitamin C and minerals using the APHA 2023 standard methods. Additionally, we will determine the product's shelf life through heterotrophic bacterial count (HBC) analysis and quarterly storage checks. Organoleptic tests will be conducted by a 10-member jury panel using a hedonic scale to score the products, and the results will be compared to standard consumer limit values. Finally, evaluate the techno-economic value of these products.

## 2. Materials and methods

### 2.1. Ethics statement

The study protocol was approved by the Ethics Committee of State University of Bangladesh (2021-01-12/SUB/A-ERC/005).

### 2.2. Sample collection and preparation

A total of 5 kilogram of ripe *Sonneratia apetala* fruits were collected from Nijum Dwip in Hatiya upazila, Noakhali district, in 11 December 2021. The study was conducted from 10 December 2021 to 09 December 2022. Four samples, each weighing 1.0 kilograms, were collected for the preparation of jam, jelly, and pickles. The GPS coordinates of the sampling point are latitude (22˚02'21.1"N) and longitude (90˚58'33.5"E). After collecting the samples, they were accurately labeled with the date, location, and collection time and then stored in a refrigerator at 4˚C. The collected samples were analyzed according to standard methods such as APHA, 2023 [12, 13].

### 2.3. Jam, jelly, and pickle processing

After harvesting, the fruits were thoroughly washed with deionized water to remove soil, sand, and dust. For the preparation of jam, jelly, and pickles, the fruits were graded, rinsed, and sorted again. The standardized method was employed for industrial-scale production of Jam, Jelly, and Pickle (Fig 1).

In a frying pan, ripe grapes were combined with NaCl salt and distilled water and fried at 80–100˚C for 5–10 minutes. After cooling to room temperature, the seeds were removed, and the mixture was blended and sieved. Sugar and water were added to the frying pan containing the blended mixture. Spices and pepper were added to the mixture for pickles. The mixture of pulp and sugar was then cooked until it reached a total soluble solids (TSS) content of 55%. At this point, the pectin-sugar mixture was added and cooking continued until the TSS reached 65% and it reached a jelly-like consistency, as measured by a refractometer. The cooking temperature was maintained at 104–105˚C. Finally, 5g of lemon juice and 0.07 gm of Sodium meta-bi-sulphide were added to each batch. The finished products were then poured into clear, dry, sterilized glass jars. The weight of the product was measured, and it was filled into glass bottles and levelled before storage, shelf life checking by quarterly storage checks, sensory analysis and marketing.

### 2.4. Quality analysis of precursor and the prepared product

**2.4.1. Proximate compositional analysis.** Using a calibrated digital electrical balance, jam, jelly, pickles, and *S. apetala* fruits weighing 15.39, 10.10, 11.13, and 12.3 grams, respectively, were carefully weighed. Samples were then dried overnight at 110˚C for 12 hours. The muffle furnace (Wise Thermo) progressively raised the temperature from 50˚C to 600˚C at six

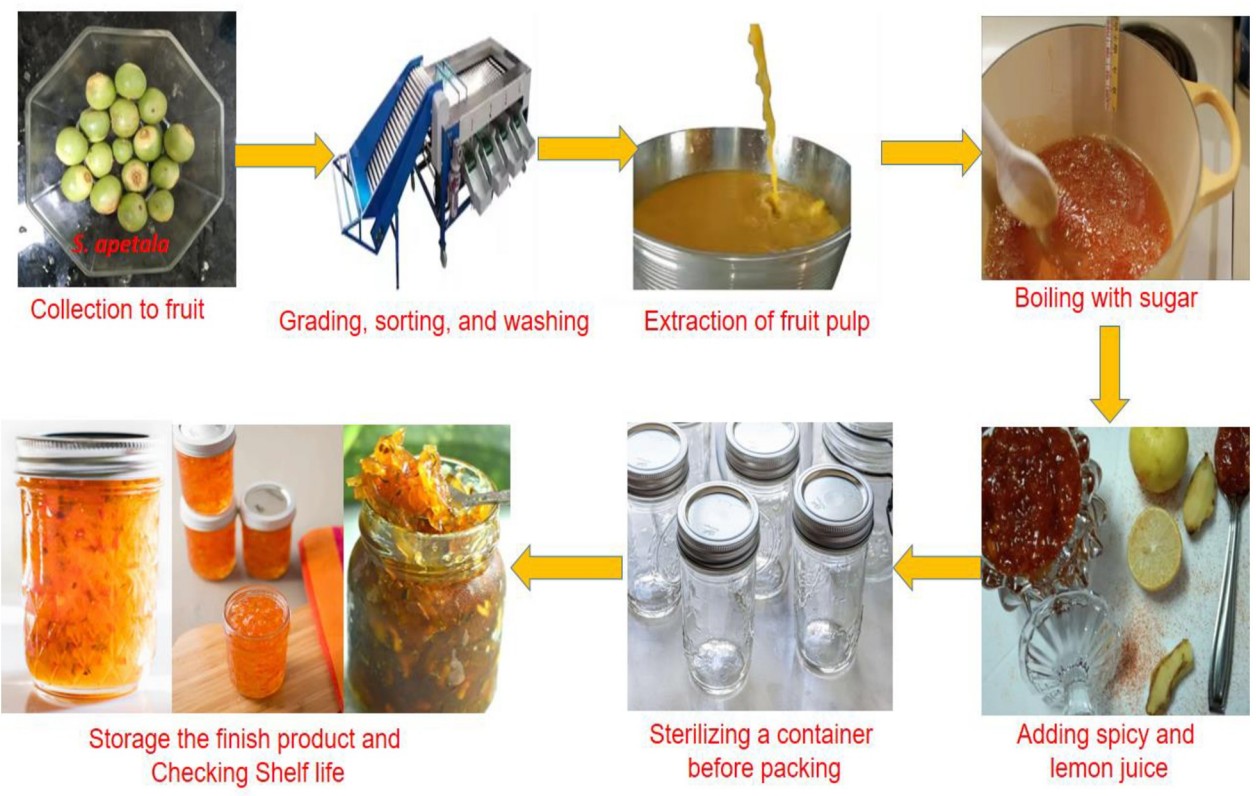

**Fig 1. Standard technique for jam, jelly, and pickle production.**

hours. Subsequently, the sample solutions were filtered to determine moisture, ash, protein, fat, fiber, carbohydrate, and energy (Kcal/kg). The methods recommended by the Association of Official Analytical Chemists [14] and the American Association for Cereal Chemists [15] were utilized for nutritional analysis.

**2.4.2. Moisture content and ash content.** Eq (1) calculates food moisture. Moisture was measured using 'A Manual of Laboratory Techniques' [14]. 5–10 grams of the sample were properly weighed into a porcelain crucible, burned to 600˚C, and then chilled to assess ash content [15].

$$\text{Ash content (g \%)} = \frac{\text{Weight of ash (g)}}{\text{Weight of sample (g)}} \times 100 \tag{1}$$

**2.4.3. Fat content, protein content, carbohydrate estimation and total energy (calorific value) determination.** The crude lipid content (%) was measured using a Soxhlet apparatus. In a Soxhlet apparatus thimble, 2 grams of sample were wrapped in Whatman filter paper (No. 1) for protection. After recording the initial weight of the Soxhlet flask, 200 ml of petroleum ether was added. The solvent was siphoned through the thimble after 8 hours of boiling at 60˚C to 80˚C. Upon completion, the flask was removed and evaporated. Eq (2) was then used, wherein the difference between the starting weight of the round joint flask and its weight after

extraction was utilized to calculate the lipid weight [14].

$$(\%) \text{ of Fat content} = \frac{\text{Weight of the fat extract}}{\text{Sample weight}} \times 100 \tag{2}$$

The Kjeldahl method, as described by [15], was employed to determine the total protein content. This method involves three steps: digestion and distillation. The crude protein value was calculated by multiplying the nitrogen percentage by 6.25, following the procedure outlined in [14].

$$\% \text{ of Nitrogen} = \frac{\text{Sample titre} \times \text{Normality of HCl} \times 14 \times \text{Sample taken} \times 100}{\text{Weight of Sample} \times 1000} \tag{3}$$

$$\text{Protein\%} = \% \text{ of nitrogen} \times \text{protein factor } (5.85) \tag{4}$$

The carbohydrate content was determined by subtracting the total contents of moisture, protein, fat, crude fiber, and ash. The energy value was calculated using the at water factor technique [(9 fat) + (4 carbs) + (4 protein)] [16]. The percentages of protein, fat, and carbohydrates were multiplied by the respective physiological fuel values.

## 2.5. Determination of nutritional compositions

The jam, jelly, pickles, and fruit samples (4.3, 5.099, 4.8, and 5.05 gm, respectively) were precisely measured using a calibrated digital electrical balance. These samples were dissolved in 250 ml volumetric flasks and filled with distilled water. The pH was determined using a pH meter (Hanna HI-255, USA) after filtering the solution. Subsequently, samplings of jam, jelly, pickles, and fruit (4.0, 3.97, 4.5, and 5.25 gm, respectively) were weighed on a calibrated digital electrical balance and dried overnight at 110°C for moisture analysis. A muffle furnace (Wise Thermo) incrementally raised the temperature to 600°C at six hours. A 2:1 mixture of concentrated $HNO_3$ and $HClO_4$ was used to produce ash samples. After practical drying at 150°C on a hotplate, 5 ml of concentrated $HNO_3$ and 25 ml of deionized water were added and heated. After cooling, the samples were diluted up to 100 ml in a volumetric flask with deionized water. Following filtering, an atomic absorption spectrophotometer (Model: AAS240FS, Varian, Australia) was utilized to measure the amounts of other metals, hazardous elements, and other metals. The UV-1650PC (Shimadzu, Japan) spectrophotometer was employed to measure total $PO_4^{3-}$. Flame photometers (Model: PFP7, Jenway, UK) were used to measure sodium and potassium concentrations. Additionally, the UV-1650PC (Shimadzu, Japan) spectrophotometer measured dissolved phosphate ions. For vitamin C analysis, a titrimetric approach [17] was employed, 3M $H_2SO_4$, one percent starch, 0.25 M vitamin C, and mixed iodine were utilized in this method. Titratable acidity was determined by acid-base method [13].

## 2.6. Quality control for elemental analysis

The samples were carefully analyzed according to ISO/IEC 17025:2017 standards. Clean pipettes, burettes, volumetric flasks, and beakers were rinsed with deionized water and a 2% (v/v) HNO3 solution before use. An electronic scale with a calibrated digital display (HR-200, Max 210 g, d = 0.1, A&D Company Limited, Japan; Model N92, D0001) were used to weigh the samples. For sample preparation, standard preparation, and instrument calibration, we were utilize deionized water (Barnstead International, USA, Model-D7071, Resistance 18.2 MΩ/cm, Conductivity 0.2 S/cm at room temperature), CRM reagents (Sigma Aldrich, Germany), and stock standards (concentration 100±4 mg/L) [12, 13]. Various checks including spike recovery tests, independent standard checks, standard checks, blank checks, and

duplicate checks were conducted during the analysis. Triplicate analyses using traceable CRM standards, samples, and reagents were performed and monitored using a quality control chart to determine analytical data accuracy and precision. Atomic Absorption Spectroscopy (AAS) and Ion Chromatography (IC) were employed to identify metals and anions up to a stock standard concentration. The detection limits for Na, Mg, K, Ca, Mn, Fe, Cu, Zn, As, Hg, Cr, Pb, and Cd are 1.0 mg/L, 0.1 mg/L, 0.1 mg/L, 0.2 mg/L, 0.005 mg/L, 0.07 mg/L, 0.05 mg/L, 0.1 mg/L, 0.01 mg/L, 0.001 mg/L, and 0.05 mg/L, respectively.

## 2.7. Evaluation of Total Soluble Solid (TSS), total sugar, reducing and non-reducing sugar

A 100 g sample of each item jam, jelly, and pickle was utilized to determine the Total Soluble Solids (TSS). The TSS was measured using an ERMA hand refractometer (Model: iTavah, COMINHKPR1216), made in Japan, which operates within a 58–92% range, at room temperature [18]. The measurement process involved pointing the refractometer's front end towards a bright light source. The diopter's adjusting ring was then fine-tuned until the reticle became clearly visible. To calibrate the device, the cover plate was opened, and one or two drops of distilled water were placed on the prism. After closing and lightly pressing the cover plate, the correcting screw was adjusted to align the light/dark boundary with the null line. Following this adjustment, the prism's surface was cleaned with soft cotton flannel. Next, one or two drops of the processed *S. apetala* product from the jam, jelly, and pickle were applied to the prism. The cover plate was closed again, pressed lightly, and the scale indicating the light and dark boundary was read. This reading indicated the TSS (measured in Brix) of the processed *S. apetala* product. These quality attributes were assessed on day 0, the day minimal processing occurred, and periodically during storage. The total, reducing, and non-reducing sugar content was measured using the Lane and Eynon method [15].

## 2.8. Bacteriological analysis

Quarterly samples of jelly, pickles, and jam were tested to determine quality. Microbiological investigation revealed possible pathogens such as *E. coli*, Vibrio, Salmonella, and Shigella, as well as the most probable number (MPN) of total and faecal coliforms. These analyses followed standard protocols. In a conical flask containing 100 ml of sterile buffered peptone water, 10 grams of each sample were suspended. Stock solutions of each sample were prepared by vigorously shaking the suspension for 5 minutes and allowing it to settle for 15 minutes to separate heavy particles. Serial dilutions ranging from 10–1 to 10–5 were then prepared from the stock solutions. 0.1 ml of each diluted sample was evenly spread onto selective media using a spread plate [19]. Culturing on nutrient agar was conducted to count heterotrophic microorganisms. For specific bacteria, *E. coli* was cultured on EMB agar, Vibrio on TCBS agar, and Salmonella and Shigella on SS agar [20, 21]. Colony forming units (CFU/gm) were counted after 24 hours of aerobic incubation at 36˚C on all plates. Following the recommendations of the (APHA, 2018), multiple-tube fermentation tests were used to count total and faecal coliforms using the Most Probable Number (MPN) method (APHA, 2018) [22]. LTB (Lauryl Tryptose Broth) and BGBB (Brilliant Green Bile Broth) were utilized for the presumptive and confirmed phases, respectively. Culture tubes were incubated for 24 to 48 hours at 37±0.2˚C for total coliforms and 44.5±2˚C for faecal coliforms when LTB medium was inoculated with samples. After observing bacterial growth, gas production, and acid-induced media color changes, the "presumptive MPN/100 gm" was determined. A negative test indicated no gas or acid generation, suggesting the absence of coliform bacteria. Presumptive LTB tubes showing growth, gas, or color changes were reinoculated in BGBB media for the confirmed phase. The final coliform

count was expressed as confirmed coliform MPN/100 gm (cultured at 37˚C) and confirmed faecal MPN/100 g (cultured at 44.5˚C). Selective media from Himedia Laboratories Ltd. were used for these tests.

## 2.9. Storage studies

Processed jam, jelly and Pickle were stored at ambient temperatures ranging from 27˚C to 34˚C for one year, during which time quality parameters such as changes in total soluble solids (TSS), Total sugar, Vit-C, reducing and Non-reducing sugar, pH, color, flavor, and texture were monitored quarterly. The analyses of these parameters were carried out according to the standard analytical methods [6, 7].

## 2.10. Sensory evaluation of jam, jelly and pickle

Sensory evaluation of the jam samples was conducted according to the methodology [23] utilizing a panel of ten members randomly selected from the Institute community. The samples were packaged in glass bottles and presented in a coded manner. The sensory attributes assessed included colour, texture, flavour, and overall acceptability. Panellists were provided with bread to accompany each coded sample and asked to grade them using a 1–9 point hedonic scale, which ranged from 'dislike extremely' to 'like extremely' for all attributes. The samples of Jam, Jelly and Pickle were stored at ambient temperature for a period of 12 months of storage and quality parameters were assessed quarterly coded with A, B, C, D, E for 0, 3, 6, 9 and 12 month respectively.

## 2.11. Statistical analysis

The data collected from storage materials and sensory analysis underwent statistical analysis were conducted using Lab Origin (Pro-9). Significant differences between mean values were assessed using p-values from two-tailed t-tests for three samples, and one-way ANOVA (Analysis of Variance) for more than two samples, with a significance threshold set at p (0.05).

## 2.12. Techno-economic evaluation for jam, jelly and pickle production

For a more comprehensive and reference-friendly formula for the techno-economic evaluation of jam, jelly, and pickle production, we can use a combination of economic indicators like Net Present Value (NPV), Internal Rate of Return (IRR), and Payback Period. General Formula for Net Present Value (NPV) [24].

$$\text{NPV} = \sum_{t=0}^{n} \frac{TRt - TCt}{(1+r)^t} - I_0 \tag{5}$$

Where: TRt = Total revenue in year t, TCt = Total cost in year t, r = Discount rate, n = Project lifespan in year, and $I_0$ = Initial investment.

$$\text{Total revenue } TRt = Pt \times Qt \tag{6}$$

Where: Pt = Selling Price per Unite in t year and Qt = Quantity sold in year t.

$$\text{Total cost } TCt = Ct + Ot + Mt + Rt \tag{7}$$

Where: Ct = Amortized capital cost in year t, Ot = Operation cost in year t, Mt = Marketing and distribution costs in year t, and Rt = Regulatory and Compliance cost in year t.

Internal rate of Return (IRR):

$$0 = \sum_{t=0}^{n} \frac{TRt - TCt - Io}{(1 + IRR)^t} \qquad (8)$$

The IRR is the discunt rate that makes the NPV of all cash flow from the project equal to zero. It helps in comparing the profitability of different investments. The Payback Period is calculated by the time it takes for the cumulative net cash flow to equal the initial investment. This measure is useful for assessing the liquidity risk of these Product.

## 3. Results and discussion

### 3.1. Mineral profile of jam, jelly and pickle

The relevance of these components is summarized in Tables 1 and 2. All products were acidic and unfavorable for microbial growth, with pH values of 3.28, 3.14, 3.02, and 2.98. The pH should not be excessively low (<3.5) as it can lead to sensory quality deterioration, characterized by glucose crystallization, granular texture, overly acidic flavor, and exudation [25]. The highest level of active acidity was observed in the pickle (0.73%), while the lowest was found in the jam (0.69%). These values indicate the presence of citric acid, which serves as an effective preservative. According to BSTI standards, the reference value for active acidity is ≤0.90%. Thus, the active acidity levels in these samples were within acceptable limits. Additionally, *S. apetala* jam, jelly, and pickle were found to be nutritionally superior to their orange and

**Table 1. Chemical and mineral profile of *Sonneratia apetala* fruits and its products.**

| Parameters | Jam (S1) | Jelly (S2) | Pickle (S3) | *S. apetala* Fruit (S4) |
|---|---|---|---|---|
| pH | 3.28 | 3.14 | 3.02 | 2.98 |
| Acidity (%) | 0.69 | 0.71 | 0.73 | 1.02 |
| TDS (mg/kg) | 19825.58 | 31378.7 | 24856.25 | 14009.9 |
| EC (μm/cm) | 40755.81 | 60600.12 | 50438.54 | 28910.89 |
| Salinity (ppt) | 0.3 | 0.6 | 0.6 | 0.2 |
| $F^-$ (mg/kg) | 95.23256 | 0 | 22.29167 | 6.930693 |
| $Cl^-$ (mg/kg) | 10873.55 | 20173.07 | 20173.44 | 3740.1 |
| $SO_4^{2-}$ (mg/kg) | 70.64 | 62.95 | 319.01 | 307.03 |
| Soluble $PO_4^{3-}$ (mg/kg) | 236.63 | 173.69 | 186.72 | 308.66 |
| Total $PO_4^{3-}$ (mg/kg) | 1385.6 | 1355.47 | 1244.44 | 1577.38 |
| $PO_4^{3-}$ as P (mg/kg) | 451.9 | 442.07 | 405.86 | 514.44 |
| Na (mg/kg) | 9873.68 | 40926.86 | 19365.9 | 2791.254 |
| K (mg/kg) | 3895.58 | 3541.81 | 3262.6 | 4267.22 |
| Ca (mg/kg) | 45.39 | 34.89 | 31.17 | 42.0 |
| Mg (mg/kg) | 18.369 | 15.25 | 8.87 | 12.6 |
| Mn (mg/kg) | 22.19 | 18.01 | 12.61 | 30.57 |
| Fe (mg/kg) | 66.96 | 67.25 | 32.17 | 75.57 |
| Cu (mg/kg) | 4.61 | 2.58 | 1.67 | 3.29 |
| Zn (mg/kg) | 10.01 | 7.03 | 5.98 | 5.9 |
| As (μg/kg) | Less than 50 | Less than 50 | Less than 50 | Less than 50 |
| Cd (μg/kg) | Less than 3 | Less than 3 | Less than 3 | Less than 3 |
| Cr (μg/kg) | Less than 50 | Less than 50 | Less than 50 | Less than 50 |
| Pb (μg/kg) | Less than 10 | Less than 10 | Less than 10 | Less than 10 |
| Hg (μg/kg) | Less than 10 | Less than 10 | Less than 10 | Less than 10 |

**Table 2. RDA values (mg/day) for different age groups of various fruit jams, jellies, and pickles.**

| Major Elements | RDA Value (mg/d) | | | | UL Value (mg/d) |
|---|---|---|---|---|---|
| | *Men* | *Women* | *Pregnancy* | *Lackting* | |
| Vit-C (mg/100gm) | 90 | 75 | 85 | 120 | 2000 |
| Na (mg/100gm) | 1500 | 1500 | 1500 | 1500 | 2300 |
| K (mg/100gm) | 2320 | 3016 | 2,500 | 2,900 | 4700 |
| Mg (mg/100gm) | 400–420 | 310–320 | 350–360 | 310–320 | 350 |
| Ca (mg/100gm) | 1000 | 1000 | 1000 | 12000 | 2500 |
| Fe (mg/100gm) | - | 18 | 27 | - | 45 |
| Mn (mg/100gm) | 2.3 | 1.8 | 2.0 | 2.6 | 11 |
| Zn (mg/100gm) | 11 | 8 | 11 | 12 | 40 |
| Cu (mg/100gm) | 0.9 | 0.9 | 1.3 | 1.3 | 10 |
| P (mg/100gm) | 700 | 700 | 700 | 700 | 4000 |
| Protein (%) | 0.8 | 1.3 | 1.8 | - | - |
| Fat (%) | 25 | 38 | - | - | - |
| Fibre (%) | - | - | - | - | - |
| Carbohydrate (%) | - | - | - | - | - |
| Energy (kcal/100 gm) | 2000 | 1600 | 500 | - | - |
| Total Suger (gm) | 25 | 36 | - | - | - |

mango counterparts [3]. We measured major and trace elements in fresh fruit pulp as well as in *S. apetala* pulp extract-based pickles, jams, and jelly.

Total Dissolved Solids (TDS) peaked at 31378.7 mg/L (Jelly) and dropped to 14009.9 mg/L (Jam). The Electrical Conductivity (EC) reached a maximum of 60600.12 μm/cm (Jelly) and a minimum of 28910.89 μm/cm (Jam). The highest Salinity was 0.6 ppt (Jelly), while the lowest was 0.2 ppt (Pulp). The highest concentrations of soluble $PO_4^{3-}$, total $PO_4^{3-}$, and $PO_4^{3-}$ as P were 308.66 mg/kg, 1577.38 mg/kg, and 514.44 mg/kg, whereas the lowest were 173.69, 1355.48, and 442.07 mg/kg. Despite its importance, excessive phosphate intake can be harmful. Overconsumption of phosphate, especially from processed foods and sodas containing phosphoric acid, may lead to cardiovascular illness, kidney difficulties, and bone loss [26].

It is crucial to maintain a balance and keep phosphate intake within daily guidelines. Health agencies such as the US Institute of Medicine (IOM) establish these Recommended Dietary Allowances (RDAs) (see Table 2). However, the adult RDA for phosphate is 700–1250 mg per day. Specifically, it is 700 mg/day for adults (19–70+) and pregnant women, and 1250 mg/day for children (9–18) [26]. Mineral phosphorus is essential for bones, teeth, and cell membranes. It activates enzymes, regulates blood pH, and is involved in the formation of DNA, RNA, and ATP, which is the body's primary energy source. Phosphorus also plays a crucial role in nerve function, muscle contraction, and heart health [26]. The RDA for phosphorus is 700 mg/day for adults, pregnancy, and breastfeeding, while the Upper Limit (UL) is 4,000 mg/day [27]. Pickle had the highest $SO_4^{2-}$ concentration at 319.01 mg/kg, while jam had the lowest at 62.95 mg/kg. Sulphate, which is essential for detoxifying drugs, dietary additives, and hazardous metals, is also known as the 'sunshine vitamin' because it depends on sun exposure, similar to vitamin D. Sulphate deficiency may lead to various health issues including autism, eczema, asthma, anemia, preeclampsia, premature delivery, and digestive problems.

The highest $F^-$ concentration was found to be 95.23 mg/kg, while the lowest value was found to be 0.0 mg/kg. Fluoride is important for dental health as it helps prevent tooth decay and strengthens tooth enamel. The RDA of Fluoride for adults and children is 0.05–0.10 mg/

kg/day, and 3–4 mg/day respectively. The UL of Fluoride for adults and children is 10 mg/day and 0.5 to 2.2 mg/day respectively. The highest $Cl^-$ concentration was found to be 10873.55 mg/kg, while the lowest value was found to be 3740.1 mg/kg. The RDA of chloride for adults and children is 2300–3600 mg/kg/day, and 1000–2300 mg/day respectively that is listed in Table 2.

In Table 1, our investigation found acceptable levels of *S. apetala* pulp extract in jam, jelly, and pickle. Arsenic (As), Chromium (Cr), Cadmium (Cd), Mercury (Hg), and Lead (Pb) were found to be absent, indicating the safety of the fruit for consumption. The human body requires small amounts of sodium (Na) to transmit nerve impulses, regulate muscle contractions, and maintain water and mineral balance [13]. For these essential functions, a daily intake of 500 mg of sodium chloride is necessary. A diet high in salt can lead to hypertension, heart disease, and stroke [13]. The Recommended Dietary Allowance (RDA) for salt is 1500 mg/day for adults, including women, pregnancy, and breastfeeding, with an Upper Limit (UL) of 2300 mg/day [28, 29]. While many foods naturally contain sodium, fruit products may also contain salt or other sodium-containing substances. For instance, *S. apetala* pulp extract jelly contains 2791.25 mg/kg of sodium, which falls within the typical range of 200 ppm (apricot) to 4800 ppm (olive pickles). Jam contained 9873.68 mg/kg of sodium, while jelly contained 19365.9 mg/kg. Potassium (K) is essential for various vital cellular functions, including maintaining blood pressure, muscle contraction, and ATP to ADP conversion [13]. Potassium RDAs are 2320 mg/day for adult men, 3016 mg/day for adult women, and 2500–2900 mg/day for pregnancy and breastfeeding, with a UL of 4700 mg/day [28, 29]. The pulp, jam, jelly, and pickles contain more potassium than sour orange juice (570 ppm). Strawberry jam contains 1407.20–1988.60 ppm of potassium. The highest potassium content observed was 4267.22 mg/kg, while the lowest was 3262.59 mg/kg.

Calcium is crucial for building strong bones, teeth, and facilitating brain-body connections [13]. It is also essential for muscular contractions, cardiovascular function, blood coagulation, cardiac rhythm modulation, and neuron activity [13]. Recommended Dietary Allowances (RDAs) for calcium are 1000 mg/day for adults, both men and women, 1000 mg/day during pregnancy, and 1200 mg/day during breastfeeding, with an Upper Limit (UL) of 2500 mg/day [28]. All samples contain 60 ppm of calcium, similar to pineapple juice. Jam contained 45.39 mg/kg of calcium, while pickle contained 31.17 mg/kg. Magnesium is essential for approximately 300 enzymes that facilitate chemical reactions in the body [13]. These reactions include protein production, maintenance of bone strength, regulation of blood sugar and pressure, and control of muscle and nerve function. RDAs for magnesium are 400–420 mg/day for adult men, 310–320 mg/day for adult women, 350–360 mg/day for pregnant women, and 310–320 mg/day for breastfeeding mothers. The UL for magnesium is 350 mg/day [28]. Jam contains 18.36 mg/kg of magnesium, while pickle contains 8.87 mg/kg.

Iron is essential for oxygen transfer in the blood, storage of myoglobin, enzyme function, and the immune system. Iron deficiency anemia can lead to fatigue and breathlessness. The Recommended Dietary Allowance (RDA) for iron is 18 mg/day for pregnant women and 27 mg/day for adult women, with an Upper Limit (UL) of 45 mg/day [30]. Copper is crucial for glucose and lipid metabolism. It assists in constructing collagen protein, energy production, bone development, and the synthesis of healthy red blood cells along with iron. The RDA of copper for adults, both men and women, during pregnancy and breastfeeding is 0.9 mg/day, with a UL of 10 mg/day [31]. Manganese, a trace mineral, is essential for human health. The RDAs for manganese are 2.3 mg/day for adult men, 1.8 mg/day for adult women, 2.0 mg/day for pregnant women, and 2.6 mg/day for breastfeeding women, with a UL of 11 mg/day [32]. Zinc, another vital trace mineral, is involved in approximately 100 enzyme-driven chemical processes, including DNA formation, cell proliferation, protein synthesis, tissue healing, and

the perception of taste and smell [32]. The RDAs for zinc are 11 mg/day for men, 8 mg/day for women, 11 mg/day for pregnant women, and 12 mg/day for nursing mothers, with a UL of 40 mg/day [32]. The order of mineral concentrations in pulp, jam, jelly, and pickle is Fe > Mn > Zn > Cu. Similarly, the concentrations of Mn, Cu, and Zn follow the order of pulp > jam > jelly > pickle. The highest concentrations observed were 30.57 mg/kg (pulp) for Mn, 75.57 mg/kg for Fe, 10.01 mg/kg (jam) for Zn, and 4.61 mg/kg (jam) for Cu, while the lowest were 12.61 mg/kg (pickle) for Mn, 32.17 mg/kg for Fe, 5.9 mg/kg for Zn, and 1.67 mg/kg for Cu.

Fig 2 presents a comparative analysis of nutritional values across different fruits and their derived products, specifically focusing on *Sonneratia apetala* fruit in the form of jam, jelly, and pickle, as compared to commonly known fruits like orange and mango. *S. apetala* fruit shows significantly higher nutrient content compared to other fruits. It has the highest Vitamin C (100.71 mg/100g), Sodium (9274.20 mg/100g), Potassium (17425.5 mg/100g), Magnesium (1440 mg/100g), Calcium (2714.29 mg/100g), Zinc (20.8 mg/100g), and Copper (11.11 mg/100g). This highlights its potential as a nutrient-dense food source. The *S. apetala* products (jam, jelly, and pickle) generally have higher levels of minerals such as Sodium, Potassium, Magnesium, Calcium, Iron, Manganese, Zinc, and Copper when compared to corresponding products made from orange and mango. Notably, the *S. apetala* jelly and jam show higher protein, fat, and carbohydrate content, indicating a more balanced nutritional profile. *S. apetala*-based products exhibit substantially higher energy and Vitamin C content, reinforcing the fruit's potential for developing high-energy, vitamin-rich food products. This suggests that *S. apetala* fruit can be utilized effectively to enhance the nutritional quality of various processed foods like jams, jellies, and pickles. The data collectively suggest that incorporating *S. apetala* fruit into different food products could provide a healthier alternative with enriched nutrients and vitamins compared to traditional fruit-based products.

To assess the nutritional value of the mangrove fruit and its by-products, we compared our data with data from common fruits (see Fig 2 and S1 and S2 Tables). The most abundant macroelement in these samples was K, followed by Na, P, Mg, and Ca. In contrast, the most prevalent trace elements were Fe, followed by Zn, Cu, and Mn (Table 2). The major mineral elements in these samples followed the order: Na > K > Ca > Mg. This suggests that these fruits could be utilized as nutritional supplements for coastal residents who may be suffering from malnutrition. Trace elements such as Zn, Cu, Mn, and Fe are believed to function as coenzymes, concentrations in food products exceeding acceptable limits may pose health risks to humans. Furthermore, it was observed that the pulp, jam, jelly, and pickles made from the fruit pulp contain trace amounts of Zn, Cu, Fe, and Mn. Specifically, the order of these elements is Fe > Mn > Zn > Cu.

## 3.2. Proximate status of jam, jelly and pickle

As shown in Table 3, the moisture content (%) of the jam, jelly, and pickle was 20.69, 15.70, and 19.59, respectively while ash content was 2.38, 3.25 and 3.26. The total sugar (%), reducing sugar (%), and non-reducing sugar (%) content of these products and fruits were 52.2, 54.3, 46.1, and 20.14; 41.9, 43.0, 36.3, and 18.04; and 10.3, 11.3, 9.8, and 2.1, respectively. From the record in Table 3 and Fig 2, it is observed that *S. apetala* fruits are rich in vitamin C, and this vitamin is also present in significant amounts in their products. The vitamin C content in fruit pulp extract, jam, jelly, and pickles is detailed in Table 3 and Fig 2. The highest vitamin C value was recorded in pulp at 9425.74 mg/kg, while the lowest was in pickle at 4958.33 mg/kg. The order of vitamin C content is Pulp > Jam > Jelly > Pickle. The vitamin C level in the pulp was 942.57 mg/100 gm but reduced to 495.83 mg/100 gm in the pickle, indicating a 19.09% reduction. This reduction may be attributed to the loss of ascorbic acid during the boiling of pulp

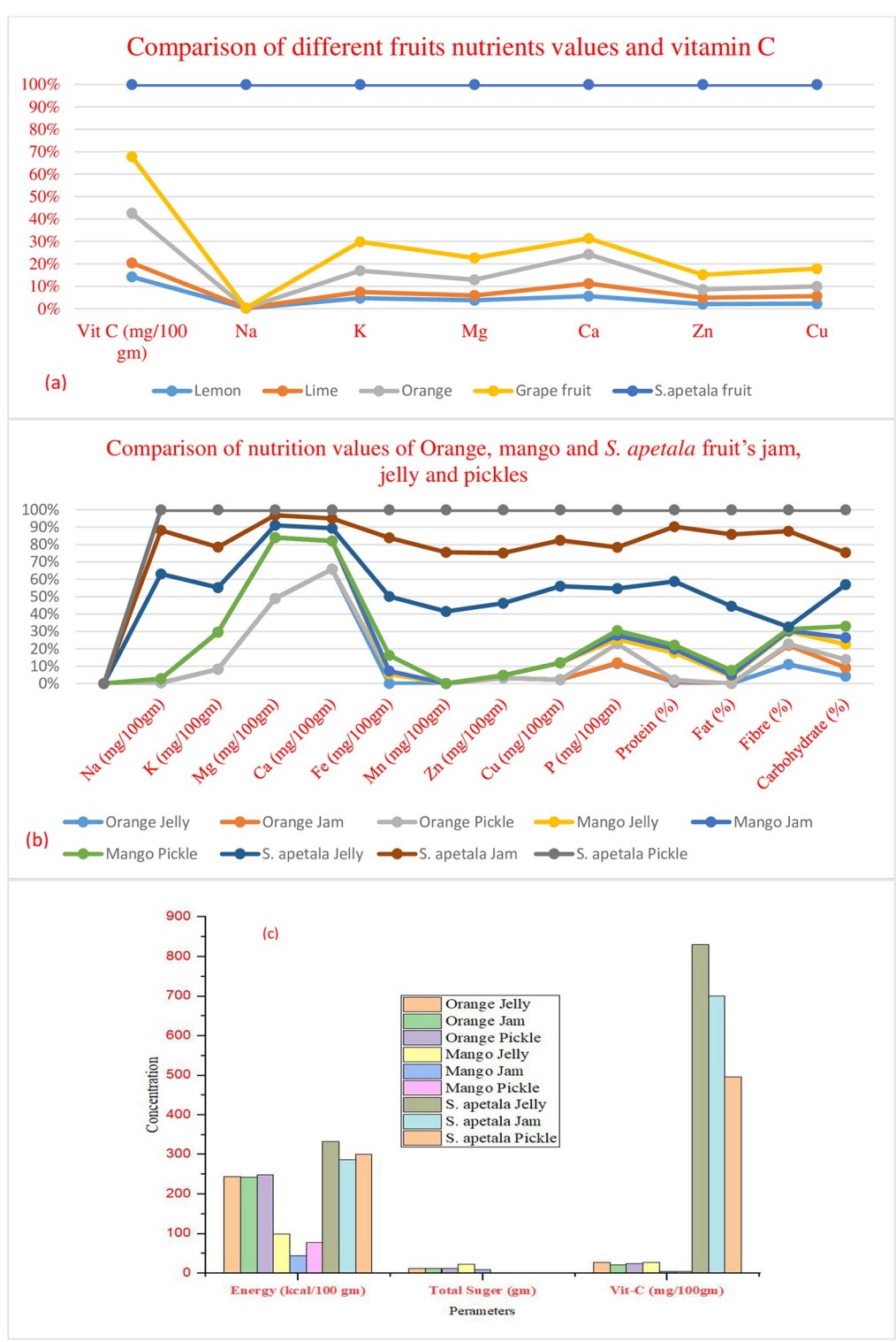

**Fig 2.** Comparison of nutrition values of (a) different fruits; (b) Orange, mango and *S. apetala* fruit's jam, jelly and pickles (c) Energy and Vit-C.

Table 3. **Proximate compositional analysis of *Sonneratia apetala* fruits and its products.**

| Parameters | Jam | Jelly | Pickle | *S. apetala* Fruit |
|---|---|---|---|---|
| Moisture (%) | 2 | 15.70 | 19.59 | 6.10 |
| Ash (%) | 2.38 | 3.25 | 3.26 | 1.35 |
| Protein (%) | 3.3 | 2.84 | 0.87 | 0.69 |
| Fat (%) | 5.36 | 5.99 | 2.05 | 1.36 |
| Fibre (%) | 0.5 | 19.98 | 4.5 | 4.94 |
| Carbohydrate (%) | 67.77 | 52.24 | 69.73 | 85.56 |
| Energy (kcal/100 gm) | 333 | 287 | 301 | 357 |
| TSS (%) | 69.2 | 68.8 | 70.5 | 40.5 |
| Total Sugar (%) | 52.2 | 54.3 | 46.1 | 20.14 |
| Reducing Sugar (%) | 41.9 | 43.0 | 36.3 | 18.04 |
| Non-Reducing Sugar (%) | 10.3 | 11.3 | 9.8 | 2.1 |
| Pectin (mg/Kg) | 3.5 | 3.6 | 3.4 | 3.5 |
| Vitamin-C (mg/kg) | 8302.3 | 7001.37 | 4958.33 | 9425.74 |

extract during pickle production, especially in the presence of seeds. Compared to some popular citrus fruits in Bangladesh like lemon, lime, orange, and grapefruit, *S. apetala* fruit contains 6–10 times more vitamin C (mg/100 gm) [28]. Vitamin C, a water-soluble antioxidant, plays a vital role in combating coughs and colds. Inadequate intake of vitamin C can lead to symptoms like anemia, scurvy, infections, bleeding gums, muscle deterioration, and delayed wound healing. The recommended daily allowances for vitamin C are 90 mg/d for adult males, 75 mg/d for adult women, 85 mg/d for pregnant women, and 120 mg/d for nursing mothers, with an upper limit (UL) of 2000 mg daily [28].

Adequate protein intake is essential for maintaining health, promoting growth, preserving muscular mass, supporting immunological function, and facilitating numerous biochemical activities in the body. Protein content ranged from 3.3% (jam) to 0.69% (pulp). Adult men and women are recommended to consume 0.8 grams of protein per kilogram of body weight per day, while pregnant women and children should aim for 1.1 to 1.3 grams [33]. The Upper Limit (UL) for protein intake in healthy individuals is currently unknown. Although overeating protein is generally considered safe for healthy individuals, excessive intake may lead to other dietary imbalances. Fat plays a crucial role in nutrient absorption, energy provision, cell structure support, and the provision of essential fatty acids. Adults should derive 20–35% of their daily calories from fat, according to the Institute of Medicine's Acceptable Macronutrient Distribution Range (AMDR). There is no established UL for total fat intake. Jelly contained 5.99% fat, while the pulp contained 1.36%. For overall health, intestinal health, blood sugar regulation, and energy provision, a balanced intake of fiber and carbohydrates from whole, nutrient-rich foods is crucial. The Institute of Medicine recommends 25 grams of fiber per day for women and 38 grams for men under 50, and 21 grams and 30 grams, respectively, for women and men over 50, on a daily basis. In jelly, 19.98% fiber was found, while in jam, it was 0.5%. The ordering of fiber content is Jelly > Pulp > Pickle > Jam. Pulp was found to contain 85.56% carbohydrate, while jelly contained 52.24%.

Calorie-based energy sustains life and supports metabolism, development, and physical activity. There is no Recommended Dietary Allowance (RDA) or Upper Limit (UL) for energy intake. Depending on individual characteristics, adult men require 2,000 to 3,000 calories per day, while women need 1,600 to 2,400 [34]. During the second and third trimesters, pregnant women may require an additional 300 to 500 calories per day compared to before. Pulp

contains 357 kcal/100 grams, while jelly contains 287 kcal/100 grams. Since sugar is not considered an essential nutrient, health authorities like protein, vitamins, and minerals have not established RDAs or ULs for sugar intake. Added sugars, especially when consumed in excess, can have detrimental effects on health. The American Heart Association (AHA) recommends limiting added sugar intake to 6 teaspoons (25 grams) per day for women and 9 teaspoons (36 grams) for men. The World Health Organization (WHO) suggests consuming less than 10% of daily calorie intake from added sugars, with further benefits observed with a reduction to below 5% of daily calorie intake. Jelly, pickle, and *S. apetala* fruits contain 3.5, 3.6, 3.4, and 3.5 grams of pectin and 520, 420, and 20 grams of sugar, respectively.

### 3.3. Shelf life status of these products

**3.3.1. Microbial status evaluation.** Based on Table 4, the total heterotrophic bacteria counts in jelly, pickle, and jam consumed over the course of a year were $32\times10^3$, $40\times10^2$, and $2\times10^2$ CFU/gm, respectively. In comparison, sealed air-free samples stored for 6 months exhibited decreased heterotrophic bacteria counts ($5\times10^2$, $2\times10^1$, and 15 CFU/gm). All samples showed total and fecal coliform counts below 1.8 MPN/100 gm. Furthermore, the processed food samples, whether consumed or stored, tested negative for *E. coli*, *Salmonella*,

**Table 4. Microbial status (shelf life) of *S. apettala* products.**

| Parameters | Code | Jam | Jelly | Pickle | (%)CV | LSD | Level of Sign. | Remarks |
|---|---|---|---|---|---|---|---|---|
| Total Heterotrophic Bacterial count log (CFU/gm) ±SD | A | 3.341±0.22 | 5.545±0.32 | 3.642±0.42 | 0.28 | 0.69 | * | Good |
| | B | 3.351±0.22 | 5.545±0.32 | 3.652±0.42 | 0.28 | 0.69 | * | Good |
| | C | 3.361±0.23 | 5.545±0.32 | 3.652±0.42 | 0.28 | 0.68 | * | Good |
| | D | 3.361±0.23 | 5.550±0.32 | 3.662±0.42 | 0.28 | 0.69 | * | Good |
| TC (MPN/100gm) | A | <1.8 | <1.8 | <1.8 | | | | Good |
| | B | <1.8 | <1.8 | <1.8 | | | | Good |
| | C | <1.8 | <1.8 | <1.8 | | | | Good |
| | D | <1.8 | <1.8 | <1.8 | | | | Good |
| TFC (MPN/100gm) | A | <1.8 | <1.8 | <1.8 | | | | Good |
| | B | <1.8 | <1.8 | <1.8 | | | | Good |
| | C | <1.8 | <1.8 | <1.8 | | | | Good |
| | D | <1.8 | <1.8 | <1.8 | | | | Good |
| *E. coli* | A | Absent | Absent | Absent | | | | Good |
| | B | Absent | Absent | Absent | | | | Good |
| | C | Absent | Absent | Absent | | | | Good |
| | D | Absent | Absent | Absent | | | | Good |
| *Salmonella* | A | Absent | Absent | Absent | | | | Good |
| | B | Absent | Absent | Absent | | | | Good |
| | C | Absent | Absent | Absent | | | | Good |
| | D | Absent | Absent | Absent | | | | Good |
| *Shigella* | A | Absent | Absent | Absent | | | | Good |
| | B | Absent | Absent | Absent | | | | Good |
| | C | Absent | Absent | Absent | | | | Good |
| | D | Absent | Absent | Absent | | | | Good |
| *Vibrio* | A | Absent | Absent | Absent | | | | Good |
| | B | Absent | Absent | Absent | | | | Good |
| | C | Absent | Absent | Absent | | | | Good |
| | D | Absent | Absent | Absent | | | | Good |

**Table 5. Storage study of *S apetala's* jam, jelly and pickle.**

| Parameters | Code | A | B | C | D | E | (%) CV | LSD value | Level of Sign. | Remarks |
|---|---|---|---|---|---|---|---|---|---|---|
| pH | Jam | 3.28 | 3.27 | 3.25 | 3.24 | 3.25 | 0.013 | 0.007 | ** | Good |
| | Jelly | 3.14 | 3.15 | 3.1 | 3.1 | 3.15 | 0.031 | 0.014 | ** | Good |
| | Pickle | 3.02 | 3.01 | 3.01 | 3.05 | 3.0 | 0.022 | 0.011 | ** | Good |
| TSS (%) | Jam | 69.2 | 69.2 | 69.2 | 69.25 | 69.25 | 0.041 | 0.020 | * | Good |
| | Jelly | 68.8 | 68.8 | 68.8 | 68.9 | 68.9 | 0.081 | 0.040 | * | Good |
| | Pickle | 70.5 | 70.5 | 70.6 | 70.7 | 70.9 | 0.129 | 0.065 | * | Good |
| Moisture (%) | Jam | 20.69 | 20.6 | 20.65 | 20.65 | 20.75 | 0.048 | 0.024 | * | Good |
| | Jelly | 15.70 | 15.6 | 15.65 | 15.75 | 15.85 | 0.029 | 0.014 | ** | Good |
| | Pickle | 19.59 | 19.5 | 19.55 | 19.6 | 19.65 | 0.290 | 0.145 | * | Good |
| Total Sugar (%) | Jam | 52.2 | 52.2 | 52.2 | 52.3 | 52.4 | 0.096 | 0.048 | * | Good |
| | Jelly | 54.3 | 54.3 | 54.4 | 54.5 | 54.5 | 0.119 | 0.059 | * | Good |
| | Pickle | 46.1 | 46.1 | 46.1 | 46.2 | 46.3 | 0.058 | 0.029 | * | Good |
| Reducing Sugar (%) | Jam | 41.9 | 41.9 | 41.9 | 42.1 | 42.2 | 0.150 | 0.075 | * | Good |
| | Jelly | 43.0 | 43.0 | 43.1 | 43.2 | 43.3 | 0.129 | 0.065 | * | Good |
| | Pickle | 36.3 | 36.3 | 36.4 | 36.4 | 36.6 | 0.126 | 0.063 | * | Good |
| Non-Reducing Sugar (%) | Jam | 10.3 | 10.3 | 10.4 | 10.4 | 10.5 | 0.058 | 0.033 | * | Good |
| | Jelly | 11.3 | 11.3 | 11.4 | 11.5 | 11.55 | 0.077 | 0.044 | * | Good |
| | Pickle | 9.8 | 9.85 | 9.9 | 9.9 | 9.95 | 0.029 | 0.017 | ** | Good |
| Acidity | Jam | 0.69 | 0.69 | 0.7 | 0.75 | 0.78 | 0.042 | 0.021 | * | Good |
| | Jelly | 0.71 | 0.71 | 0.72 | 0.73 | 0.73 | 0.009 | 0.005 | ** | Good |
| | Pickle | 0.73 | 0.73 | 0.74 | 0.74 | 0.75 | 0.008 | 0.004 | * | Good |
| Vit-C (mg/kg) | Jam | 8302.3 | 8302.1 | 8302 | 8302 | 8302 | 0.05 | 0.025 | * | Good |
| | Jelly | 7001.37 | 7001.37 | 7001.31 | 7001.32 | 7001.32 | 0.027 | 0.014 | ** | Good |
| | Pickle | 4958.33 | 4958.33 | 4958.31 | 4958.3 | 4958.3 | 0.014 | 0.007 | ** | Good |

*Shigella*, and *Vibrio*. For microbial preferences, the one-way ANOVA showed no significant differences ($p < 0.05$). Despite the benefits of *S. apetala*-processed foods, ensuring quality and standards is imperative.

Food microbiological quality is commonly assessed using the heterotrophic bacterial count (HBC), which influences product shelf life. As a food product approaches its expiration date, HBC tends to increase. In this investigation, the HBC records for *S. apetala* jelly, pickle, and jam were consistent with previous studies. Given their low pH and high sugar content, jam, jelly, and pickle create a hypertonic environment that deprives microbes of the water they need to survive. The International Commission on Microbiological Specifications for Foods recommends a range of $10^2$ to $10^6$ CFU/gm for fruit-derived foods, depending on the samples and storage conditions. However, the absence of *total coliforms*, *total fecal coliforms*, *E. coli*, *Vibrio sp.*, *Salmonella sp.*, and *Shigella sp.* indicates superior and safer fruit products. However, their presence suggests issues with sanitation, non-hygienic packing materials, and post-processing contamination due to poor handling and hygiene.

**3.3.2. Storage data evaluation.** Table 5 reveals that sample E exhibited the highest moisture content, with 20.75% for jam, 15.85% for jelly, and 19.65% for pickle. In contrast, sample B recorded the lowest moisture content, with 20.6% for jam, 15.6% for jelly, and 19.5% for pickle. Moisture content is a critical indicator of the shelf life of food products. For instance, researcher observed a reduction in the moisture content of dragon fruit jelly from 28.90% to 27.15% over 90 days at room temperature, with a slight decrease to 27.56% under refrigeration [35]. Similarly, research reported that the moisture content of guava-carrot jelly decreased

from 24.02% to 21.58% during storage [36] and found a decline in karonda jelly's moisture content from 35.44% to 30.87% [37]. Furthermore, study noted that the moisture content of apple jam dropped from 76.99% to 75.33% over a 28-day storage period [38]. These findings underscore the significance of monitoring moisture content to ensure the quality and longevity of preserved food products. Among the three samples, sample A exhibited the highest pH levels, with 3.28 for jam, 3.14 for jelly, and 3.02 for pickle, while sample E recorded the lowest pH levels, with 3.25 for jam, 3.14 for jelly, and 3.0 for pickle. Conversely, sample E had the highest titratable acidity, with 0.78% for jam, 0.73% for jelly, and 0.75% for pickle, whereas sample A had the lowest values, with 0.69%, 0.71%, and 0.73%, respectively. Study reported a decrease in the pH of dragon fruit jellies over four months of storage [39]. This reduction in pH across all treatments could be attributed to the degradation of ascorbic acid or the hydrolysis of pectin [40] in diet apple jam, [41] in apple and pear mixed fruit jam, and [42] in wood apple jelly.

Table 5 indicates that sample A had the highest vitamin C content, with 8302.3 mg/L for jam, 7001.37 mg/L for jelly, and 4958.33 mg/L for pickle. In contrast, sample E had the lowest vitamin C levels, with 8302.0 mg/L for jam, 7001.32 mg/L for jelly, and 4958.3 mg/L for pickle. Ascorbic acid content typically decreases during preservation due to degradation by anti-ascorbic compounds. In jam, this reduction may also result from the oxidation of ascorbic acid to dehydroascorbic acid (DHA), furfural, or hydroxymethyl furfural. These observations are consistent [43], who reported an 18.79% decrease in the ascorbic acid content of jackfruit jam after six months of storage. Similarly, study [10] suggested that the reduction in ascorbic acid content in wood apple jelly could be due to oxidation from trapped oxygen in glass bottles, leading to the formation of volatile and unstable dehydroascorbic acid, which further degrades into 2,3-diketogulonic acid and eventually furfural compounds.

An increase in Total Soluble Solids (TSS) indicates a reduction in moisture content, thereby enhancing the nutritional quality of the jam samples. Sugars and fruit acids are the primary contributors to TSS in fruit-based products. Generally, a higher TSS value signifies a greater sugar content in the sample. In our study, the highest TSS levels were observed in sample E, with 69.25% for jam, 68.9% for jelly, and 70.9% for pickle, while the lowest levels were 69.2%, 68.8%, and 70.5%, respectively. Pectins and metal salts (such as sodium, potassium, and calcium) can also slightly influence TSS values. According to Desrosier and Desrosier, the standard TSS range should be slightly above 65% [30]. The Codex Alimentarius standard (CODEX STAN, 2009) states that typical fruit preserves should contain at least 60% soluble solids; for instance, the TSS of melon jam was found to be 73%. An increase in TSS may result from the acid hydrolysis of polysaccharides, particularly pectin and gums. Additionally, the solubilization of ingredients during storage can elevate TSS levels in jams. In contrast, study found [44] no significant change in the TSS of mixed fruit marmalades over a six-month storage period. Study reported [10] a gradual increase in the TSS of wood apple jelly over the storage period.

Table 5 reveals that sample E had the highest levels of reducing sugar, with 42.2% for jam, 43.3% for jelly, and 36.6% for pickle. In contrast, sample A had the lowest levels, with 41.9%, 43.0%, and 36.3%, respectively. Sample E also exhibited the highest total sugar content, with 54.4% for jam, 54.5% for jelly, and 46.3% for pickle, while sample A had the lowest, with 54.2%, 54.3%, and 46.3%, respectively. Similarly, the highest non-reducing sugar content was found in sample E, with 10.5% for jam, 11.55% for jelly, and 9.95% for pickle, while the lowest was in sample A, with 10.3%, 11.3%, and 9.8%, respectively. Previous research indicates that reducing sugar content tends to increase during storage due to the acidic nature of the samples [33]. Literature underscores the importance of maintaining a balance between sucrose and invert sugar ratios in jams and jellies [33, 34]. It is recommended to limit invert sugar content, with a higher proportion of sucrose. A previous study suggested that reducing sugar content

**Table 6. Sensory evaluation of jam, jelly, and pickle across four quartiles (one year).**

| Parameters | Code | A | B | C | D | E | (%) CV | LSD value | Level of Sign. | Remarks |
|---|---|---|---|---|---|---|---|---|---|---|
| Color | Jam | 7.50 | 7.45 | 7.40 | 7.4 | 7.45 | 0.047 | 0.024 | * | Good |
| | Jelly | 7.8 | 7.75 | 7.7 | 7.7 | 7.75 | 0.047 | 0.024 | * | Good |
| | Pickle | 8.1 | 8.0 | 8.05 | 8.0 | 8.0 | 0.047 | 0.025 | * | Good |
| Flavor | Jam | 7.60 | 7.55 | 7.50 | 7.45 | 7.45 | 0.065 | 0.032 | * | Good |
| | Jelly | 7.9 | 7.85 | 7.8 | 7.75 | 7.70 | 0.065 | 0.032 | * | Good |
| | Pickle | 8.15 | 8.1 | 8.05 | 8.0 | 8.0 | 0.065 | 0.032 | * | Good |
| Texture | Jam | 7.30 | 7.35 | 7.40 | 7.35 | 7.3 | 0.04 | 0.021 | * | Good |
| | Jelly | 7.65 | 7.65 | 7.7 | 7.65 | 7.6 | 0.025 | 0.012 | * | Good |
| | Pickle | 8.15 | 8.05 | 8.0 | 8.05 | 8.0 | 0.062 | 0.031 | * | Good |
| Overall Acceptability | Jam | 7.40 | 7.45 | 7.40 | 7.45 | 7.40 | 0.021 | 0.014 | * | Good |
| | Jelly | 7.75 | 7.70 | 7.75 | 7.7 | 7.7 | 0.021 | 0.014 | * | Good |
| | Pickle | 8.05 | 8.0 | 8.0 | 8.05 | 8.0 | 0.021 | 0.014 | * | Good |

The means with same superscripts within a column are not significantly different at p<0.05.

should range between 20–40% to prevent crystal separation during storage [27]. Recent formulations incorporating glucose syrups in jams and jellies have significantly reduced crystal formation [7]. A one-way ANOVA analyzing pH, acidity, vitamin C, moisture, TSS, total sugar, reducing sugar, and non-reducing sugar preferences showed no significant differences ($p < 0.05$). These findings indicate that the product is of good quality and safe for consumption.

## 3.4. Organoleptic status evaluation

Jam, jelly, and pickle samples were subjected to sensory evaluation by 10 judges, who rated color, flavor, texture, and overall acceptability for three samples. All samples were made with the same ingredients, and mean scores for these attributes are presented in Table 6. Jam, jelly, and pickle samples were evaluated by a panel of 10 judges for color, flavor, texture, and overall acceptability. Each sample was made with the same ingredients, and the mean scores for these attributes are presented in Table 6. The one-way ANOVA showed no significant differences ($p < 0.05$) in preferences for color, flavor, texture, and overall acceptability, indicating no significant difference in organoleptic preferences. As shown in Table 6, Sample A was the most preferred for color, followed by Sample B, with Samples C, D, and E being equally acceptable at a 0.05 significance level.

Notably, Sample A received the highest color scores, with 7.5 for jam, 7.8 for jelly, and 8.1 for pickle out of 9. Sample A was also the most favored overall, scoring 7.6 for jam, 7.9 for jelly, and 8.15 for pickle out of 9. For texture, Sample A for pickle and Sample C for jam and jelly were the most preferred, with the highest scores being 7.40 for jam, 7.7 for jelly, and 8.15 for pickle out of 9. Table 6 indicates that Sample A had the best overall acceptability, scoring 7.45 for jam, 7.75 for jelly, and 8.05 for pickle out of 9. Sensory analysis provides marketers with insights into product quality, directions for enhancing product attributes, and facilitates profiling competing products and evaluating product reformulations from a consumer perspective [45]. The primary quality attributes valued by consumers include sensory characteristics such as texture, flavor, aroma, shape, and color. Researcher observed [46] a declining trend in the overall acceptability scores of banana-pineapple blended jam over the storage period, attributing this decline to the deterioration in color, texture, taste, and flavor as storage time increased.

## 3.5. Techno-economic status evaluation

The market prices of fruit jams, jellies, and pickles from popular brands vary slightly when converted to US dollars. For fruit jams, brands like Kissan, Himalaya, and Mapro generally range between $1.80 to $3.00 per 500g, depending on the brand and packaging size. Jellies from brands such as Kissan, Mala's, and Mapro are priced between $1.50 to $2.40 per 500g. Meanwhile, pickles from well-known brands like Mother's Recipe, Pachranga, and Priya typically cost around $1.00 to $2.20 for packs ranging from 300g to 500g. These prices are approximate and may vary based on the location, store, and additional factors such as import duties and local taxes. The global market for jam, jelly, and pickles demonstrates significant demand, market value and medicinal value [47]. According to Market Research BIZ, these products reached a market size of $1.7 billion in 2022, with an annual growth rate of 5.5%. It is projected that by 2032, the market will expand to $2.9 billion. Annually, ten thousand metric tons of *S. apetala* fruit are harvested along the coastal belt and in the Sundarbans. From one kilogram of *S. apetala* fruit, the final yields are 1 kilogram of jam, 1 kilogram of jelly, and 1.5 kilograms of pickles. The production period is limited to two months each year. To operate the 1.75 kW automatic jam, jelly, and pickle production machine, one technician and three laborers are needed daily. This machine has a production capacity of 4,000 bottles per hour (BPH). Using Eqs 1–4, we can calculate the Net Present Value (NPV) for a jam, jelly, and pickle production business. According to these equations, the Initial Capital Costs (Ct) are $10,000, amortized over 10 years. The Annual Operational Costs (Ot), Annual Marketing and Distribution Costs (Mt), and Annual Regulatory Costs (Rt) are $5,000, $1,000, and $1,000, respectively, totaling $17,000 in Annual Total Costs (TCt). The finished product's average Selling Price per Unit (Pt) is $10, with an Annual Quantity Sold (Qt) of 10,000 units. This results in a Total Annual Revenue (TRt) of $100,000. The investment payback period is two years, with the project's lifespan limited to 10 years and a discount rate of 8%.

Scaling up the production of jams, jellies, and pickles from *S. apetala* fruits poses significant environmental concerns, particularly regarding the sustainability of wild harvesting practices. Increased demand could lead to overharvesting, threatening local biodiversity, disrupting ecological balance, and reducing the availability of fruits for wildlife. To ensure the sustainability of harvesting *S. apetala* fruits from the wild, a combination of regulated harvesting practices, community-based management, and habitat conservation is essential. Implementing guidelines that limit the volume and frequency of fruit collection, promoting the cultivation of *S. apetala* in controlled environments, and engaging local communities in sustainable practices can help maintain ecological balance. Additionally, monitoring fruit populations and restoring degraded habitats would support long-term sustainability and biodiversity preservation.

Harvesting *Sonneratia apetala* presents several potential ethical issues, particularly concerning its impact on local ecosystems and the traditional uses of the fruit by indigenous populations. Large-scale harvesting could disrupt local biodiversity, leading to habitat loss for species that rely on these trees, and potentially altering the ecological balance. Additionally, *Sonneratia apetala* may hold cultural, medicinal, or economic significance for indigenous communities, and commercial exploitation could infringe on their traditional rights and practices. To address these concerns, it is crucial to implement sustainable harvesting practices that involve local communities, ensuring that their cultural and environmental interests are protected and that they benefit equitably from any commercial ventures.

## 4. Conclusions

In the modern age, people are increasingly conscious of their health and dietary choices. They seek out foods that are not only delicious but also safe, of high quality, low in fat, with an

optimal balance of nutrients, fewer calories, and less sugar. Additionally, consumers look for enticing options that are reasonably priced. These requirements can be met by utilizing *Sonneratia apetala* fruit to produce Jam, Jelly, and Pickles. Our project aims not only to assess the nutritional value of these fruit products but also to offer alternative means of sustenance to the island communities living along Bangladesh's coast. *Sonneratia apetala* stands out for its excellent protein, fiber, carbohydrate, and energy content, making it a mineral-rich and healthy fruit option for people. Moreover, these fruits and their products are free from toxic substances, and their nutrient content stays within safe limits for consumers. They also boast outstanding mineral and vitamin C levels when compared to other citrus fruits. The products have a shelf life of up to one year, thanks to their lower pH value and a significant presence of organic acid (vitamin C), which acts as a natural food preservative. The absence of *total coliform*, *total fecal coliform*, *E. coli*, *Vibrio sp.*, *Salmonella sp.*, and *Shigella sp.* ensures superior and safer quality in these fruit products. However, it's essential to recognize that food safety is a critical factor in addressing hunger and malnutrition among the delta's population. *Sonneratia apetala* can play a crucial role in addressing regional food security, basic healthcare, income generation, salinity management, climate change mitigation, and can be cultivated in vast saline areas beyond coastal embankments. Jam, jelly, and pickled foods are known for their antioxidant, antibacterial, analgesic, antidiarrheal, anti-diabetic, depressive, and cytotoxic properties. Today, home-based businesses have the opportunity to produce jam, jelly, fruit bars, and marmalade. These industries flourish through preservation and promotion efforts. With proper attention and investment, this underutilized indigenous fruit can contribute significantly to our economy by creating job opportunities for islanders and small to medium-sized businesses. These new small-scale mangrove-based enterprises require collaboration between research units, academic institutions, island communities, and businesses.

We suggested the need for further research involving a larger, more diverse panel that reflects various demographic and cultural backgrounds to better understand the products' acceptance, explore the applicability of the findings to a broader range of settings, helping to better understand how these variables might affect the results and preference across different consumer segments. In the future, exploring modern, more efficient methods such as high-pressure processing (HPP), vacuum evaporation, and advanced packaging technologies could significantly enhance the quality and shelf life of *S. apetala* jams, jellies, and pickles. These methods preserve nutritional value and sensory qualities by minimizing thermal degradation while reducing microbial load, thereby extending shelf life and maintaining product freshness. Further studies could provide a more detailed understanding of how these factors affect nutrient absorption from *S. apetala*-based products. Allergen testing and risk assessments specific to jam, jelly, and pickle products derived from *Sonneratia apetala* is necessity of conducting comprehensive allergen tests, such as skin prick tests, IgE binding assays, and oral food challenges, to identify any potential allergens present in the fruit or other plant parts used in these products. Additionally, it will address the importance of performing risk assessments that evaluate the likelihood of allergic reactions among different consumer groups, including those with common fruit or tree nut allergies. The section will also cover labelling requirements, such as including allergen warnings on packaging, to inform consumers and minimize health risks.

## Supporting information

**S1 Table. Comparison of different fruits nutrients values and vitamin C.**
(DOCX)

**S2 Table. Comparison of nutrition values of orange, mango and *S. apetala* fruit's jam, jelly and pickles.**
(DOCX)

## Acknowledgments

The authors are grateful to the authority of the Institute of National Analytical Research and Service (INARS), BCSIR, Dhaka, 470 Bangladesh and Ministry of Science and Technology, The People 's Republic of Bangladesh for providing analytical, technical, and other logistic support for conducting this research work.

## Author Contributions

**Conceptualization:** Md. Ripaj Uddin.

**Data curation:** Md. Ripaj Uddin, Md. Abu Bakar Siddique.

**Formal analysis:** Md. Ripaj Uddin, Md. Abu Bakar Siddique, Shahnaz Sultana, Umme Hafsa Bithi.

**Investigation:** Md. Ripaj Uddin, Shahnaz Sultana, Umme Hafsa Bithi, Nahida Akter, Abubakr M. Idris, Muhammad Abdullah Al Mansur, AHM Shofiul Islam Molla Jamal.

**Methodology:** Md. Ripaj Uddin, Umme Hafsa Bithi, Nahida Akter, Abubakr M. Idris, Muhammad Abdullah Al Mansur, AHM Shofiul Islam Molla Jamal.

**Project administration:** Md. Ripaj Uddin.

**Supervision:** Md. Ripaj Uddin.

**Visualization:** Md. Ripaj Uddin.

**Writing – original draft:** Md. Ripaj Uddin.

**Writing – review & editing:** Md. Ripaj Uddin, Nahida Akter, Abubakr M. Idris, Mayeen Uddin Khandaker.

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
