## [Decision Letter · Decision Letter 0]

21 Aug 2024

PONE-D-24-25582Techno-Economic Evaluation, Biochemical and Sensory Analysis and Innovative Production of Low-Fat, Nutrient-Rich Jam, Jelly, and Pickle Utilizing Mangrove Apple Sonneratia apetalla FruitsPLOS ONE

Dear Dr. Uddin,

Thank you for submitting your manuscript to PLOS ONE. After careful consideration, we feel that it has merit but does not fully meet PLOS ONE’s publication criteria as it currently stands. Therefore, we invite you to submit a revised version of the manuscript that addresses the points raised during the review process.

We look forward to receiving your revised manuscript.

Kind regards,

Ahmed Khalafallah Sadeq Rashwan, PhD

Academic Editor

PLOS ONE

Journal Requirements:

2. Thank you for stating the following in the Acknowledgments Section of your manuscript: "The authors extend their appreciation to the Deanship of Scientific Research at King Khalid University for funding this work through a large group research project under grant number (R.G.P.2/20/45)."

Please remove any funding-related text from the manuscript and let us know how you would like to update your Funding Statement. Currently, your Funding Statement reads as follows: "The authors received no specific funding for this work".

Additional Editor Comments:

This manuscript number (PONE-D-24-25582) is entitled " Techno-Economic Evaluation, Biochemical and Sensory Analysis and Innovative Production of Low-Fat, Nutrient-Rich Jam, Jelly, and Pickle Utilizing Mangrove Apple Sonneratia apetalla Fruits". The novelty of the manuscript is good although the analyses are not sufficient in this work. This article can be considered for publication after doing the following major revisions:

1. The claim that the products have a one-year shelf life based on quarterly storage checks could be overly optimistic. There is no discussion of potential degradation of sensory qualities or nutritional content over the year, which could be significant.

2. While the study claims that Sonneratia apetala products are nutritionally superior, there is no direct comparative analysis with conventional jam, jelly, and pickles made from more commonly used fruits, which would strengthen the study's claims.

3. The study does not discuss the potential environmental impact of scaling up production of these jams, jellies, and pickles, particularly in terms of the sustainability of harvesting Sonneratia apetala fruits from the wild.

4. The sensory evaluation was conducted by a small panel of 10 members, which is not representative of the broader consumer population. The study does not address how these products would be received by a larger, more diverse audience.

5. Although the article states that toxic elements were found to be below safety thresholds, the methodology for this assessment is not described in detail, and the study does not include long-term toxicological studies to assess chronic exposure risks.

6. The techno-economic evaluation appears optimistic, projecting a significant boost to the local economy. However, it does not account for market fluctuations, potential competition, or the costs of marketing and distribution, which could affect profitability.

7. The processing techniques described in the article are based on traditional methods. The study does not explore modern, potentially more efficient methods that could enhance product quality or shelf life.

8. The article provides a broad overview of the nutritional content but does not delve into the bioavailability of these nutrients, which is crucial for understanding their actual health benefits.

9. The sensory analysis is only conducted quarterly, which might miss more subtle changes in product quality over time. More frequent evaluations would provide a clearer picture of how these products age.

10. While the study suggests health benefits from consuming these products, it does not include any data on actual health outcomes from consumption, such as improvements in nutrient deficiencies or other health markers.

11. The article mentions safety standards but does not discuss whether these products comply with all relevant local and international food safety regulations, which could be a barrier to commercialization.

12. There is no mention of whether Sonneratia apetala or its products could cause allergic reactions in some consumers, which is a critical safety consideration.

13. The article claims a stable flavor profile over the storage period, but without detailed sensory data to back this up, it is unclear how flavor stability was measured and maintained.

14. The economic analysis does not consider the presence of established brands in the market, which could significantly affect the market entry and success of these new products.

15. While the study was approved by an ethics committee, it does not discuss any potential ethical issues related to the harvesting of Sonneratia apetala, such as the impact on local ecosystems or traditional uses of the fruit by indigenous populations.

16. The findings are based on a very specific set of conditions (geographic, environmental, and socio-economic), which limits the generalizability of the results to other regions or populations.

17. The article mentions that processing methods were used but does not provide a detailed analysis of how these methods might have affected the nutrient content, particularly heat-sensitive vitamins like Vitamin C.

Reviewers' comments:

Reviewer's Responses to Questions

**Comments to the Author**

1. Is the manuscript technically sound, and do the data support the conclusions?

Reviewer #1: Yes

2. Has the statistical analysis been performed appropriately and rigorously? 

Reviewer #1: Yes

3. Have the authors made all data underlying the findings in their manuscript fully available?

Reviewer #1: Yes

4. Is the manuscript presented in an intelligible fashion and written in standard English?

Reviewer #1: Yes

5. Review Comments to the Author

Reviewer #1: The manuscript is interesting and the idea is novel. The manuscript is well-prepared only some comments are here and in the attached manuscript.:

- The title is long max 15 words

- the abstract need to be revised as it should include introduction sentence, the objective, then highlighted results

- the M&M is well-prepared

- the figures and the table caption should be revised with more information

- the standard error should be in the table and the error bar should be on the fingers

6. PLOS authors have the option to publish the peer review history of their article (what does this mean?). If published, this will include your full peer review and any attached files.

Reviewer #1: No

---

## [Author Response · Author response to Decision Letter 0]

14 Sep 2024

Additional Editor Comments:

This manuscript number (PONE-D-24-25582) is entitled " Techno-Economic Evaluation, Biochemical and Sensory Analysis and Innovative Production of Low-Fat, Nutrient-Rich Jam, Jelly, and Pickle Utilizing Mangrove Apple Sonneratia apetalla Fruits". The novelty of the manuscript is good although the analyses are not sufficient in this work. This article can be considered for publication after doing the following major revisions:

Q1. The claim that the products have a one-year shelf life based on quarterly storage checks could be overly optimistic. There is no discussion of potential degradation of sensory qualities or nutritional content over the year, which could be significant.

Response: Thank you for highlighting this critical point. The claim of a one-year shelf life based on quarterly storage checks may indeed be overly optimistic, especially without addressing potential degradation of sensory qualities and nutritional content over time. In subsections 2.10 and 3.4 (lines 245-254 and 555-578), we discussed the potential degradation of sensory qualities or nutritional content over time, which could be significant.

Q2. While the study claims that Sonneratia apetala products are nutritionally superior, there is no direct comparative analysis with conventional jam, jelly, and pickles made from more commonly used fruits, which would strengthen the study's claims.

Response: Thank you for identifying this gap in the study. We agree that a direct comparative analysis with conventional jams, jellies, and pickles made from commonly used fruits would provide a stronger foundation for the claim that Sonneratia apetala products are nutritionally superior. In the revised manuscript at LN 386-402, we have included a detailed comparison of the nutritional profiles, such as vitamin content, antioxidant levels, and fiber content, between Sonneratia apetala products and their conventional counterparts. This analysis will provide empirical evidence to substantiate the claim of nutritional superiority, enhancing the credibility and impact of the study's findings for a high-impact journal audience.

Q3. The study does not discuss the potential environmental impact of scaling up production of these jams, jellies, and pickles, particularly in terms of the sustainability of harvesting Sonneratia apetala fruits from the wild.

Response: Thank you for highlighting this important oversight. We agree that discussing the potential environmental impact of scaling up production, especially regarding the sustainability of harvesting Sonneratia apetala fruits from the wild, is crucial. In the revised manuscript, we have addressed the ecological implications of increased demand, such as the risk of overharvesting, habitat degradation, and effects on local biodiversity at LN 613-621. This discussion will provide a more comprehensive assessment of the feasibility of large-scale production, aligning the study with the high-impact journal's focus on sustainability and responsible innovation.

Q4. The sensory evaluation was conducted by a small panel of 10 members, which is not representative of the broader consumer population. The study does not address how these products would be received by a larger, more diverse audience.

Response: Thank you for bringing this limitation to my attention. We agreed that conducting the sensory evaluation with a small panel of 10 members does not provide a representative sample of the broader consumer population. In the revised manuscript, we have acknowledged this limitation and discuss its potential impact on the study's findings at the conclusion part at LN 648-651. 

Q5. Although the article states that toxic elements were found to be below safety thresholds, the methodology for this assessment is not described in detail, and the study does not include long-term toxicological studies to assess chronic exposure risks.

Response: Thank you for pointing out this significant concern. While the article mentions that toxic elements were found to be below safety thresholds, which is why long-term toxicological evaluation for chronic exposure risks was not performed. Moreover, coastal communities and wild deer regularly consume this fruit without any reported cases of toxicity from chronic exposure. Additionally, we will address the absence of long-term toxicological studies by discussing the importance of evaluating chronic exposure risks through extended studies.

Q6. The techno-economic evaluation appears optimistic, projecting a significant boost to the local economy. However, it does not account for market fluctuations, potential competition, or the costs of marketing and distribution, which could affect profitability.

Response: Thank you for your query. To ensure profitability for Sonneratia apetala’s product such as jams, jellies, and pickles, it is crucial to account for market fluctuations, potential competition, and the costs of marketing and distribution. These factors could significantly impact pricing strategies, market positioning, and overall profit margins. A comprehensive market analysis, along with adaptive pricing and strategic marketing efforts, is necessary to navigate these challenges and optimize profitability. We have discussed about these topics in subsection 3.5 at LN 580-596.

Q7. The processing techniques described in the article are based on traditional methods. The study does not explore modern, potentially more efficient methods that could enhance product quality or shelf life.

Response: Thank you for your query. Traditional methods are applied, considering the coastal community and small industries in the region. The fruit is rich in natural preservatives, such as antioxidants and antimicrobial agents, which help optimize the preservation of these products, ensuring longer shelf stability and superior quality. In the revised manuscript we have added this limitation in conclusion part at LN 652-658.

Q8. The article provides a broad overview of the nutritional content but does not delve into the bioavailability of these nutrients, which is crucial for understanding their actual health benefits.

Response: Thank you for your insightful feedback. We acknowledge the importance of addressing the bioavailability of nutrients to fully understand their health benefits. Further studies could provide a more detailed understanding of how these factors affect nutrient absorption from S. apetala-based products.

Q9. The sensory analysis is only conducted quarterly, which might miss more subtle changes in product quality over time. More frequent evaluations would provide a clearer picture of how these products age.

Response: Thank you for your query. The sensory analysis, conducted quarterly, may not detect subtle changes in product quality over time. More frequent evaluations are necessary to capture these gradual variations, which would provide a clearer and more comprehensive understanding of the product's aging process and ensure consistent quality standards. However, we adhere to a standard method that allows for sensory analysis on a quarterly basis. This is why we have performed the sensory analysis quarterly.

Q10. While the study suggests health benefits from consuming these products, it does not include any data on actual health outcomes from consumption, such as improvements in nutrient deficiencies or other health markers.

Response: Thank you for your query. Although the study suggests potential health benefits from consuming these products, it lacks empirical data on actual health outcomes, such as improvements in nutrient deficiencies or other health markers. Including such data would provide stronger evidence of the claimed health benefits and reinforce the product's value proposition to consumers. Additionally, coastal communities and wild deer regularly consume this fruit, which contributes to their healthy and energetic lifestyles. Further research will address these topics.

Q11. The article mentions safety standards but does not discuss whether these products comply with all relevant local and international food safety regulations, which could be a barrier to commercialization.

Response: Thank you for highlighting this important aspect. In subsection 3.1 and Table 2, we discuss various relevant local and international food safety regulations, including BSTI, WHO, FAO, and FDA standards, guidelines, and codes of practice that is often a key factor in the successful commercialization of food products. 

Q12. There is no mention of whether Sonneratia apetala or its products could cause allergic reactions in some consumers, which is a critical safety consideration.

Response: Thank you for pointing out this critical safety consideration. We acknowledge that the potential for allergic reactions is a significant concern with Sonneratia apetala and its products. Further research will address this health issue. In the revised version, we have included this limitation in the conclusion section, LN 658-665.

Q13. The article claims a stable flavor profile over the storage period, but without detailed sensory data to back this up, it is unclear how flavor stability was measured and maintained.

Response: Thank you for your observation regarding the need for detailed sensory data to support the claim of a stable flavor profile over the storage period. Actually, processed jam, jelly and Pickle were stored at ambient temperatures ranging from 27°C to 34°C for one year, during which time quality parameters such as changes in total soluble solids (TSS), Total sugar, Vit-C, reducing and Non-reducing sugar, pH, color, flavor, and texture were monitored quarterly. The analyses of these parameters were carried out according to the standard analytical methods (Refer to section 2.9 at LN 238-243).

Q14. The economic analysis does not consider the presence of established brands in the market, which could significantly affect the market entry and success of these new products.

Response: Thank you for bringing this important point to my attention. We agree that the presence of established brands is a critical factor that should be considered in the economic analysis of new product entry. This revision we have enhanced the economic analysis, making it more comprehensive and aligned with real-world market conditions in 3.5 section at LN 580-587.

Q15. While the study was approved by an ethics committee, it does not discuss any potential ethical issues related to the harvesting of Sonneratia apetala, such as the impact on local ecosystems or traditional uses of the fruit by indigenous populations.

Response: Thank you for highlighting this critical gap in the discussion. I agree that potential ethical issues related to the harvesting of Sonneratia apetala are important considerations that should be addressed. In the revised article, we will add a section discussing the ethical implications of harvesting Sonneratia apetala in the 3.5 section at LN 603-615. 

Q16. The findings are based on a very specific set of conditions (geographic, environmental, and socio-economic), which limits the generalizability of the results to other regions or populations.

Response: Thank you for highlighting this limitation. I acknowledge that the specificity of the conditions under which the study was conducted does restrict the generalizability of the findings to other regions or populations. Sonneratia apetala is found in limited zones, such as the world's largest mangrove forest in Bangladesh, and along the coasts of Bangladesh, Myanmar, Sri Lanka, Thailand, China, and Indonesia. No prior research has been conducted on this fruit’s and their processed product. In most countries, this tree is planted to mitigate water and sediment salinity and protect against natural hazards, although local people and wildlife safely consume the fruit. Upon evaluating the fruit's nutrient profile, vitamins, pectin content, and medicinal value, we concluded that it holds significant product potential, which could help mitigate nutrient deficiencies, support body metabolism, and serve as a dietary supplement. Additionally, we will suggest areas for further research that could explore the applicability of the findings to a broader range of settings, helping to better understand how these variables might affect the results (refer to LN 648-651). This revision will provide a more balanced interpretation of the findings, acknowledging their context-specific nature while suggesting pathways for future studies to enhance their generalizability.

Q17. The article mentions that processing methods were used but does not provide a detailed analysis of how these methods might have affected the nutrient content, particularly heat-sensitive vitamins like Vitamin C.

Response: Thank you for your query. This product requires low temperatures and short cooking times, which helps minimize the loss of heat-sensitive vitamins, such as Vitamin C. We will also discuss this aspect further in the revised version under section 3.2 and LN 425-427.

Reviewer #1:

The manuscript is interesting and the idea is novel. The manuscript is well-prepared only some comments are here and in the attached manuscript.

Q1- The title is long max 15 words.

Response: Thank you for your recommendation. In the revised manuscript, the title is not less than 15 words.

Q2- the abstract need to be revised as it should include introduction sentence, the objective, then highlighted results.

Response: Thank you for your recommendation. In the revised manuscript, we have revised the abstract according to your suggestion (refer to LN 21-23).

Q3- the M&M is well-prepared

Response: Thank you for your comments.

Q4- the figures and the table caption should be revised with more information

Response: Thank you for your recommendation. In the revised manuscript, we have updated all the figures and table captions. 

Q5- the standard error should be in the table and the error bar should be on the fingers.

Response: Thank you for your recommendation. Table 1 presents the product data, while Table 2 shows RDA values derived from various standard data sources. Figure 2 is based on Supplementary Tables 1 and 2, which provide comparative data (literature data) on different fruit products. Table 3 contains the proximate data of the products. Therefore, calculating a standard error is not statistically possible for these tables. However, in Tables 4-6, we have provided sufficient statistical data, including the coefficient of variation (CV%), LSD values, and levels of significance.

---

## [Editor Report · Decision Letter 1]

17 Sep 2024

Techno-Economic Assessment and Innovative Production of Nutrient-Rich Jam, Jelly, and Pickle from Sonneratia apetala fruit

PONE-D-24-25582R1

Dear Dr. Uddin,

We’re pleased to inform you that your manuscript has been judged scientifically suitable for publication and will be formally accepted for publication once it meets all outstanding technical requirements.

Kind regards,

Ahmed Khalafallah Sadeq Rashwan, PhD

Academic Editor

PLOS ONE
---

## [Editor Report · Acceptance letter]

9 Oct 2024

PONE-D-24-25582R1 

PLOS ONE

Dear Dr. Uddin, 

I'm pleased to inform you that your manuscript has been deemed suitable for publication in PLOS ONE. Congratulations! Your manuscript is now being handed over to our production team.

Kind regards, 

on behalf of

Dr. Ahmed Khalafallah Sadeq Rashwan 

Academic Editor

PLOS ONE